# Resistance Welding of Thermoplastic Composites, Including Welding to Thermosets and Metals: A Review

**DOI:** 10.3390/ma17194797

**Published:** 2024-09-29

**Authors:** Karolina Stankiewicz, Adrian Lipkowski, Piotr Kowalczyk, Maciej Giżyński, Bartłomiej Waśniewski

**Affiliations:** Lukasiewicz Research Network—Institute of Aviation, 02-256 Warsaw, Poland; adrian.lipkowski@ilot.lukasiewicz.gov.pl (A.L.); piotr.kowalczyk@ilot.lukasiewicz.gov.pl (P.K.); maciej.gizynski@ilot.lukasiewicz.gov.pl (M.G.); bartlomiej.wasniewski@ilot.lukasiewicz.gov.pl (B.W.)

**Keywords:** fiber-reinforced plastic composite, thermoplastic resin, thermosetting resin, adhesion, resistance welding, joints/joining, dissimilar materials joining

## Abstract

This review paper presents the current progress in the development of resistance welding techniques for thermoplastic composites, with a particular emphasis on their application in hybrid joints, such as those involving thermosetting composites and metals. Resistance welding, a fusion bonding method, offers significant advantages over adhesive bonding and mechanical joining by eliminating the need for additional adhesive materials and enabling integration into automated manufacturing processes. The study highlights the unique benefits of resistance welding, including lower energy consumption compared to other methods and its compatibility with automated manufacturing, which can reduce production costs by up to 40%. Key findings from the literature indicate that resistance welding is particularly effective in achieving strong, durable joints for complex and large structures, such as those used in the aerospace industry. The review also identifies the main challenges associated with resistance welding, including temperature control, current leakage in carbon-fiber-reinforced polymers, and potential corrosion when using metal meshes. To address these challenges, various strategies are discussed, including surface treatments, the use of nanocomposites, and the integration of carbon nanotubes. The review concludes by emphasizing the need for further research to optimize welding parameters and to develop non-destructive testing methods for industrial applications, ensuring the reliability and long-term performance of welded joints.

## 1. Introduction

A significant challenge emerging with the rising interest in fiber-reinforced thermoplastic polymer (FRTP) composites in industries such as aerospace and automotive is the process of joining composite components with other composites or different groups of materials, including metals. Unlike traditional thermoset composites, which cannot be joined through fusion due their irreversible curing process, thermoplastic composites can be effectively fused because of their ability to be remelted. Thermoset composites, while highly resistant to chemicals and heat, have the limitation of not being reconfigurable or recyclable once cured. Their primary advantage is their superior strength, making them highly effective for structural applications where long-term environmental stability is required, such as in aerospace and defense applications [1,2,3,4,5]. However, their long processing times and irreversible nature make them more expensive for large-scale manufacturing applications compared to thermoplastics [6,7].

In addition to their fusion bonding capabilities, FRTP composites generally offer higher production rates due to reduced cycle times and the potential for post-forming and reconfiguration, along with the ability to operate at higher temperatures, particularly in composites like polyphenylene sulfide (PPS), polyetherimide (PEI), polyether ether ketone (PEEK), and polyaryletherketone (PAEK). These materials can withstand high-temperature environments where traditional thermosets might degrade [8]. Although thermoset composites are known for their excellent thermal and chemical resistance, making them ideal for specific high-performance applications, FRTP composites are gaining popularity due to their faster production, recyclability, and ability to be reshaped and reused in different processes, reducing overall production costs in industries focused on efficiency and sustainability [7,9].

While the initial material costs for FRTPs may be higher than those for thermosets, the overall production costs can be lower due to faster processing times and the potential for recycling and reconfiguration, providing long-term economic benefits [10]. Additionally, thermoplastic prepregs do not require refrigerated storage and have an essentially unlimited shelf life, further reducing logistical and storage costs compared to thermoset composites [11,12]. In comparison, thermoset composites require more complex storage and handling processes, such as refrigeration, to prevent premature curing, adding logistical challenges [13]. Recent research, such as [14,15,16,17,18,19,20,21,22,23,24,25,26,27,28], has further explored these material’s properties and potential applications.

In the assembly of large composite structures, established industry practices involve the use of mechanical fasteners (such as bolts and rivets) as well as bonding techniques. Mechanical fasteners offer several advantages, such as eliminating the need for surface preparation and facilitating the easy disassembly of components. However, when subjected to loads, mechanically fastened joints can experience micro-damage in the fasteners and holes due to the undesirable accumulation of stress, leading to structural deterioration of the joint. Moreover, mechanical fasteners present challenges for composite materials due to their distinct thermal expansion properties, added weight, and the need for additional manufacturing processes. There is also a risk of water infiltration, delamination at the joint interface, fiber fracture, and other issues [29,30,31,32].

Adhesive joints involve the joining of two components by applying an adhesive between them. With advancements in polymer bonding technology, adhesives have seen improvements in terms of strength, fatigue life, and stiffness. The primary advantage of using adhesive joints is their ability to bond various materials while ensuring limited stress concentration along the joint line [33,34]. However, adhesive joints also have several drawbacks. They require longer surface preparation and curing times compared to mechanical fasteners. Additionally, the bond strength of adhesive joints is not as robust as that of mechanical fasteners, and they lack the capability for disassembly [35]. The prolonged curing times associated with bonding methods pose a significant challenge in manufacturing processes, making the search for alternative, quicker joining techniques appealing in serial production.

Another option for joining thermoplastic composites is the fusion bonding process, which involves heating and melting the polymer on the joining surfaces of the components and subsequently pressing these surfaces together to solidify and consolidate the polymer. Fusion bonding techniques exploit the flow properties of thermoplastic matrices when heated above their glass transition temperature (T_g_) for amorphous polymers or melting temperature (T_m_) for semi-crystalline polymers. These techniques are typically categorized based on the heating method employed. Heat can be delivered to the interface through ultrasound, radio frequency and microwave signals, induction, hot plate, hot gas, or laser technology [36]. Among the various methods, three have shown significant promise: induction welding, ultrasonic welding, and resistance welding. Additionally, other notable techniques include laser-assisted welding, conduction welding, and friction welding [37,38,39,40,41,42,43,44,45,46]. Each of them is optimized for specific applications and geometries for components commonly found in the aerospace, space, and automotive sectors.

Induction welding is particularly effective for continuous weld lines on large, flat, or moderately contoured surfaces of CFRP, offering rapid and uniform heating, though it necessitates conductive materials and entails higher initial setup costs [47,48].

Ultrasonic welding is best suited for small welds. Although traditionally limited to spot welds, research at TU Delft is exploring continuous ultrasonic welding, which could significantly expand its application, potentially enabling long, continuous welds suitable for large structures like fuselage joints [49,50].

Resistance welding is highly effective for joining thermoplastic composites due to its ability to generate precise, localized heat at the weld interface, ensuring strong, durable joints. It is versatile, capable of handling a variety of materials, and can accommodate different part thicknesses, ranging from 3 mm to 30 mm. It allows for complete control over the temperature at the joint interface and the pressure applied during welding, which is essential for scalability in larger applications. A key factor in the process is the precise management of temperature ramps, which are critical for certain thermoplastic polymers. Proper control of temperature helps prevent material degradation and ensures that the polymers melt and bond without affecting the structural integrity. Additionally, the positioning of both elements to be joined is carefully managed, ensuring proper alignment and optimal bonding conditions. While the process does leave the resistive element embedded in the final part, requiring careful thermal management—particularly in thicker components—the method remains highly suitable for large, complex structures. This is exemplified by its application in the Airbus A320 rear pressure bulkhead, where long, continuous welds are required for structural integrity [48].

Given the challenges associated with achieving high-quality bonded joints for thermoplastic composites [51], there is a natural progression toward advancing welding methods for this material group. Among the various methods—ultrasonic, resistance, and induction—it has been demonstrated that resistance welding has the lowest electricity consumption during the process [52].

Resistance welding offers the significant advantage of integrating fabrication and joining into automated manufacturing processes, ensuring the ability to seamlessly join thermoplastic composites with other automated manufacturing methods, potentially reducing production costs by up to 40% [53,54]. However, further research is needed to fully confirm these savings under industrial-scale conditions. Resistance welding technology has been increasingly adopted in aerospace applications, as evidenced by its use in the design of Airbus aircraft components, with studies conducted by GKN Fokker and DLR [8,55]. DLR reports that a weld length of up to 1.500 mm can be effectively achieved in a single joining process. In specific cases, it is even possible to consider welded joints as detachable, though this requires additional planning during the joint design stage [56]. Current research efforts focus on characterizing and certifying resistance welded joints, particularly for aerospace applications [57].

This work provides a comprehensive review of resistance welding techniques for thermoplastic composites, with a focus on their application in joining dissimilar materials such as thermosetting composites and metals. The study systematically explores the principles and processes involved in resistance welding, highlighting key factors that influence weld quality, such as the choice of heating elements, process parameters, and the thermal management of the materials being joined. By analyzing recent advancements and research findings, the paper underscores the growing significance of resistance welding in industries like aerospace, where it is increasingly utilized for constructing large and complex composite structures. Additionally, the work discusses the challenges associated with resistance welding, including temperature control, potential material degradation, and current leakage, while presenting strategies like surface treatments and the use of nanocomposites to enhance weld quality. This review also addresses the need for further research to optimize welding parameters and develop reliable non-destructive testing methods, ensuring the long-term performance and industrial applicability of welded joints in high-performance composite materials.

## 2. Resistance Welding

Resistance welding is a widely used technique for joining materials, particularly in thermoplastic composites, by generating heat through electrical resistance to melt the material at the interface of a joint. This process primarily relies on the principles of Joule heating, where the electrical energy is converted into thermal energy as current passes through a resistive element, causing localized heating at the interface. The process begins when an electric current is applied to the heating element situated between two components. This heating element can be a metallic mesh, carbon fibers, or a nanocomposite, each chosen based on its specific resistive properties. The flow of electric current through this element elevates its temperature to Joule’s law (as shown in Equation (1)):(1)$$Q=I^{2}\cdot R\cdot t$$

Here, *Q* is the heat generated, *I* is the current, *R* is the resistance of the element, and *t* is the time for which the current is applied.

Figure 1 shows a schematic of the resistance welding system. The basic setup for resistance welding consists of an isolated pressing block, an isolated base plate, a heating element, electrodes, and a current source.

As the temperature at the interface rises, it causes the polymer around the heating element to melt. This melting is crucial for achieving a strong bond, as it allows the polymer chains to flow and intermingle at the molecular level under the applied pressure, solidifying to form a welded joint upon cooling.

The resistance welding process typically involves several key stages:Heating: The components are aligned and brought into contact under pressure. An electric current is applied, causing the resistive heating element to heat. The temperature at the interface begins to rise. As the temperature increases, the polymer reaches its melting point, allowing it to flow and wet the interface.Consolidation: The current is maintained for a set duration, allowing for complete melting and fusion at the interface.Cooling and solidification: The electric current is carefully adjusted to control the cooling rate of the composite. This adjustment ensures that the material cools at an appropriate rate, depending on the polymer matrix used in the composite [58,59,60,61,62].

The thermal process, consisting of these three stages, is schematically represented in the Figure below (Figure 2). The diagram illustrates a process where constant pressure is applied throughout each stage. However, an alternative approach involves applying lower pressure during the initial stages (heating and early consolidation) and increasing it during the later stages of consolidation and cooling [63].

### 2.1. Heating Element

A resistive element typically refers to a component of welding, such as metallic meshes [64,65,66,67,68,69,70,71,72,73], carbon fibers [74,75,76], or nanocomposite heating element [77,78], all of which generate heat during the process. These materials can effectively serve as heating elements. However, carbon fibers and nanocomposite heating elements offer the added advantage of becoming a mono-material with the laminate, meaning they can integrate seamlessly with the composite material, enhancing the overall welding process. This integration is particularly effective when the matrix of the nanocomposite is made from the same thermoplastic polymer as the welded component, ensuring a homogeneous bond between the materials.

When selecting the appropriate heating element, it is crucial to consider the material properties and the desired outcome of the final product. Ensuring that the heated material remains free of undesirable impurities is especially important. The heating element’s material, such as steel mesh or carbon fibers, directly affects the thermal and electrical conductivity, which in turn influences the quality and uniformity of the joint. While steel meshes provide rapid heating, they are prone to corrosion [79], particularly in high-humidity environments or in the presence of aggressive chemicals, potentially compromising the joint’s long-term durability. In such cases, corrosion-resistant alternatives like carbon fibers may be more suitable, although their conductive properties differ from those of steel [80].

Elements made of carbon fiber or metal mesh have distinct thermal and electrical conductivity properties. Specifically, the meta mesh functions as a hot-carrier thermistor (Negative Temperature Coefficient, NTC), meaning its resistance decreases as temperature increases. In contrast, the carbon fiber heating element acts as a cold-conductor (Positive Temperature Coefficient, PTC), where its resistance increases with rising temperature. These differing characteristics significantly impact temperature control: the NTC metal mesh allows for rapid heating and efficient thermal distribution, while PTC carbon fiber ensures stable temperature regulation by self-limiting the heat generation as temperatures rise. Examples of the electrical and thermal properties of steel mesh and carbon fiber are shown in Table 1.

### 2.2. Process Parameters

Several parameters affect the quality and efficiency of welds, including input power, thermal insulation, welding temperature and time, pressure, and, as mentioned above, the type of heating element [83]. Table 2 summarizes these parameters and their effects, which influence both the welding process and the quality of the weld.

Thermal analysis using finite element modelling (FEM) has been identified as a crucial method for assessing the thermal insulation, energy input, and heating duration necessary for adequate interface heating, which is essential for achieving a high-quality welded joint. Effective thermal insulation, combined with sufficient energy input, can significantly enhance the welding process by reducing the required welding time and simultaneously improving the quality of the weld [25].

Tanaka [85] observed in his study that applying high pressure (2.0 MPa) during resistance welding of carbon fiber-reinforced polyamide 6 (CF-PA6) leads to resin extrusion outside the welding zone, resulting in low shear strength. Conversely, a thesis by Huajie Shi conducted at Delft University of Technology [86] noted that high pressures (1.5 MPa) can be used to reduce voids formed by residual volatile substances within the composite, such as moisture and N-Methyl-2-pyrrolidone, a solvent used in the impregnation process.

The duration of heat application (holding time) in fusion bonding of thermoplastic composites is crucial for achieving optimal weld strength [85,87]. Adequate holding time allows for sufficient melting and intermingling of the polymer chains at the welding interface, leading to stronger bonding. Tanaka [85] observed in his research that the optimal welding time for CF-PA6 is 30 s. However, over longer durations (60, 120 s), the surface of specimens was melted and damaged by the heat transfer from the heating element.

The temperature used in the resistance welding of thermoplastic composites has a significant impact on the process and the quality of the weld [87]. Higher welding temperatures can improve the flow of the thermoplastic matrix, enhancing fusion between the materials. However, excessively high temperatures may degrade the thermoplastic, depending on the thermoplastic matrix used. While resistance welding typically does not generate an open flame, the localized heat can affect both the thermoplastic and thermosetting composites. Particular caution is required when joining high-performance thermoplastic composites with thermosetting composites, as thermosets generally have lower operating temperatures than high-performance thermoplastics [88]. When joining materials with different softening or melting points, it is essential to carefully control heat input to prevent degradation of the material with the lower thermal threshold [78,89]. Dual polymer joints are commonly used in processes like Thermabond™ [89], where amorphous polymer layers are co-molded onto laminates. In this process, the heating element is typically made from the same material as the laminates, usually a semi-crystalline polymer. A crucial aspect of this technique is ensuring that the glass transition temperature (T_g_) of the parent material is higher than the melting temperature (T_m_) of the co-molded polymer layer, allowing the welding to occur without compromising the integrity of the laminates [90].

When joining two different thermoplastic composites, compatibility between the polymers is critical. This compatibility helps to preserve the inherent properties of both materials during the welding process, resulting in a durable and high-quality joint [89]. Careful control of welding parameters is particularly important when the materials have different softening or melting points, to avoid thermal degradation of the more sensitive material.

## 3. Thermoplastic Composite to Thermoplastic Composite Joints

Resistance welding is used for joining thermoplastic composites because it offers several advantages over adhesive bonding and mechanical fastening. Adhesive bonding can be challenging because thermoplastic materials are generally harder to bond compared to thermosets due to their lower surface free energy. This makes the surface less chemically reactive and less conducive to good adhesion. Additionally, the lack of a rigid cross-linked network in thermoplastics reduces the effectiveness of surface preparation techniques like abrasion [91,92,93,94,95]. Moreover, adhesive bonding is often time-consuming [95,96]. In contrast, resistance welding provides a faster, more reliable method by precisely heating the weld interface, ensuring strong, durable bonds without the need for additional materials or the stress concentrations that can occur with mechanical fasteners.

As mentioned in the previous section, in the welding of high-performance thermoplastic composites, carbon fibers (CF) prepreg (fabric or unidirectional) can be stacked as a heating element (HE) to form joints without any foreign materials at the interface, thereby increasing the long-term integrity of the joint [74,75,76,85,97]. Joints resistance welded using fabric heating elements exhibited higher lap shear strength (LSS) (up to 69% improvement) and a higher Critical Strain Energy Release Rate—G_Ic_ (up to 179% improvement) compared to those welded using unidirectional heating elements. Additionally, using fabric heating elements for resistance welding proved to extended the processing window to longer processing times [97].

To ensure a smooth contact surface and to minimize the negative impact of trapped voids inside the welded joint, the heating element should be fully consolidated. To facilitate autoadhesion of the polymer–polymer interfacial surface, it is advantageous to provide a resin-rich layer on the bonding surface. For this reason, the heating element can be placed between two layers of thermoplastic films of the polymer included in the composites to be bonded before consolidation [64,67,71]. It has been noted that the shear strength of CF-PEI specimens welded using a heating element (CF-PEI) with a polyetherimide (PEI) film is up to 30% higher than that of specimens fabricated with a heating element without PEI film [76].

The main advantage of using CF as a heating element is its compatibility with the reinforcing material in welded composites. However, it should be noted that the anisotropic thermal and electrical properties of unidirectional CFs and fabrics provoke difficulties in controlling the temperature distribution during welding process [97,98].

In contrast, using a metal mesh (e.g., stainless steel) as a heating element enables a more uniform temperature distribution in the welding area, resulting in a weld with good coherence and high strength [35,64,69,70,71,99,100,101]. The metal mesh also makes the weld less sensitive to changes in welding parameters, thereby widening the processing window. The production process resembles that of CF heating elements and laminates, where the mesh is co-formed between layers of polymer, providing a rich resin content that reduces air entrapment between the wires [70,99]. Research has shown that PEI-based laminates can be satisfactorily bonded using a heating element with a stainless steel (SS) mesh, achieving Lap Shear Strength (LSS) equivalent to that of reference samples. Fracture analysis revealed excellent adhesion between the mesh and the polymer matrix [99].

Resistance welding is a suitable method for joining thermoplastic composites reinforced with glass fibers due to the insulating nature of the glass material. However, welding carbon-fiber-reinforced thermoplastic polymer (FRTP) composites is more challenging due to current leakage into the carbon reinforcement. Various methods have been considered to insulate the metal wire mesh, but the most reliable approach has been to use two layers of glass-fiber fabric for insulation [74,75]. This method was successfully applied in the resistance-welded main landing gear doors of the Fokker 50 in the 1990s [102].

Recent studies have suggested that resistance welding, using a welding element with a carbon fiber conductor without glass insulation, instead employing a neat resin film, may also be promising option. This approach can further minimize the presence of foreign materials within the welded seams [74].

The use of metal mesh in resistance welding may lead to corrosion issues, particularly when combined with semi-crystalline matrices. In such cases, the metal implant can compromise the matrix’s inherent corrosion resistance. Therefore, special attention must be given to sealing the joint against corrosive environments. Additionally, the metal mesh may introduce extra weight, create stress concentrations, and pose challenges related to differing thermal expansion between the mesh and the components being joined [35].

It has been observed that enhancing the mechanical properties of an implant consisting of a metal mesh and thermoplastic film positively impacts the weld joint strength [71]. Therefore, surface modification of the metal heating element may prove advantageous in improving the interfacial bonding between the metal mesh and thermoplastic resin, and consequently, the quality and performance of the implant.

To improve the interface between the stainless steel mesh heating element and the thermoplastic polymer polyphenylene sulfide (PPS), researchers [64] applied a silane coating to the stainless steel mesh. Welded connections with the coated heating element exhibited significantly better adhesion between stainless steel and PPS. The joints welded with untreated and coated heating elements had an average LSS of 28.5 ± 1.7 and 37.6 ± 2.2 MPa (an increase of 32%), respectively. Additionally, the developed silane coating can prevent moisture absorption at the SS/PPS interface, which is significant in the context of corrosion [64].

In a similar approach, other researchers employed chemical etching of the stainless steel wire surface [71]. The mesh was first treated with 7 mol/L hydrochloric acid (HCl) to increase the surface roughness of the stainless steel wire wires. The etched mesh was then placed between two PEI films for resistance welding GF-PEI. The etching pits on the on the stainless-steel surface significantly enhanced the bonding strength with the resin. Consistent with this observation, the surface roughness increased with the etching time, improving adhesion to the PEI polymer. When the etching time and welding times were 30 min and 150 s, respectively, LSS reached a maximum of 35.44 MPa, a27.7% joints with untreated SS mesh [71]. However, LSS values began to decline when the etching time exceeded 30 min due to the mechanical degradation of the stainless steel.

The literature also reports on the use of nanocomposites as a heating element or adhesion enhancers [78,103,104,105]. Introducing 10% by weight of multi-walled carbon nanotubes (MWCNTs) into the PEI matrix provided the nanocomposite with electrical conductivity (0.79 Scm^−1^), which successfully enabled the welding of CF-PEEK laminates, achieving average LSS ranging from 13 MPa to 16.6 MPa [78]. Some authors [78] suggest that using the nanocomposite as HE can reduce thermal stresses, as the thermal expansion coefficient of MWCNT/PEI is better matched to that of CF/PEEK composites compared to pure PEI with metallic insertion.

During the optimization process of welding GF-PEI joints, research has focused on a heating element reinforced with graphene oxide (GO), referred to as GO-PEI [104]. The results indicated that an optimal GO content of 0.2% by weight significantly improved the LSS of the welded joints. This optimal concentration increased the LSS to 39.5 MPa, representing a 35.8% enhancement. However, this type of heating element requires further research due to the inhomogeneous temperature distribution in the weld area.

The integration of carbon nanotube (CNT)-reinforced steel mesh as heating elements has enhanced welded connections and interface strength [103]. CNTs improve the wettability of polyetherimide on the SS mesh, leading to stronger material bonding. Additionally, CNTs create more robust connections between the resin and SS grid, increasing structural integrity and thermal efficiency.

When considering the use of nanomaterials, it is crucial to address the challenges associated with their industrial-scale application [106,107]. While producing nanomaterials in a laboratory setting is relatively straightforward, scaling up these processes to an industrial level presents significant technological and industrial challenges. Ensuring the uniformity of nanomaterials in large production batches is difficult. Inconsistent dispersion of nanoparticles within the polymer can lead to unpredictable properties in the final material.

Research in the literature explores resistance welding of high-performance thermoplastic composites, including carbon and glass fiber-reinforced polyetherimide (CF-PEI, GF-PEI), carbon and glass fiber-reinforced polyphenylene sulfide (CF-PPS, GF-PPS), carbon fiber-reinforced polyether ether ketone (CF-PEEK), carbon fiber low-melt poly aryl ether ketone (CF-(LM-PAEK)), and carbon fiber-reinforced polyether ketone ketone (CF-PEKK). Selected examples are compiled in Table 3.

Beyond the laboratory scale, there is significant interest in resistance welding of thermoplastic composites for larger structures. This reflects the growing focus within the scientific community on applying this welding technique to more substantial and complex constructions, expanding research and application beyond initial small-scale experiments. Notably, resistance-welded main landing gear doors on the Fokker 50 and j-nose leading edge wing structures on the Airbus A340 and A380 have been successfully in service for decades [55].

The Multifunctional Fuselage Demonstrator (MFFD), funded by the Clean Sky 2 program, represents a significant advancement in thermoplastic composites welding. This A320-type fuselage section, measuring 4 m in diameter and 8 m in length, will be the world’s largest thermoplastic composite structure once its upper and lower halves are welded together. The lower shell, developed by the STUNNING consortium, and the upper shell, constructed by a consortium including DLR, Premium Aerotec, Aernnova, and Airbus, were welded together by Fraunhofer in Stade. Resistance welding was used in the upper shell for joining frames to skin and cleats (shear ties) to stringers. The DLR utilized a carbon heating element for resistance welding to avoid introducing foreign materials like stainless steel mesh in the joint, ensuring uniform material in the weld [74,75].

In the “LuFoV-3 TB-Rumpf” project, DLR and Airbus collaborated to showcase an integrated aircraft fuselage section. This section featured a skin/stringer panel, which was curved and consolidated using an out-of-autoclave (OOA) vacuum process and was constructed with resistance-welded frames and cleats. The materials used for this demonstration included CF-PAEK UD tape [55,109].

The resistance heating process developed under the “Repair Technology for Thermoplastic Aircraft Structures” (REPTAS) program by Northrop Corporation is a proven method for on-aircraft repairs [12]. The repair begins with surface preparation of the PEEK structure, requiring etching and priming, as direct co-consolidation of PEI onto PEEK is not feasible. A repair patch made from APC-2 PEEK, with a co-consolidated PEI film layer, is prepared using an autoclave at 0.2 MPa and 385 °C for 30 min. The patch is then bonded to the aircraft using resistance heating in two stages: first, heating to 150–165 °C for 60 min to remove moisture, followed by increasing the temperature to 285–330 °C for 30 min to complete the bond. The use of an amorphous PEI layer enables bonding above the glass transition temperature of PEI but below the melting point of PEEK, maintaining the integrity of the structure. This method ensures a durable bond, free from internal stress or delamination, and was shown to hold even when the structure failed at 115% of the design ultimate load. The process is suitable for field-level repairs with minimal training required.

Beyond the aerospace industry, which has shown a keen interest in fusion bonding of thermoplastic composites using resistance welding, the wind energy and automotive sectors are also exploring this technology [110]. As shown in Figure 3, panels of 1 m^2^ and a 5-meter-long blade tip were manufactured using the thermoplastic resin Elium^®^ (Arkema, France), featuring fusion-welded joints and lightening protection system [111].

This chapter extensively details how resistance welding leverages the distinct thermal and electrical properties of materials such as carbon fiber and metal mesh to create strong, durable bonds. The chapter also highlights the use of innovative materials like carbon nanotubes and graphene oxide to further enhance the welding process.

The challenges of resistance welding, including temperature control, current leakage in carbon-fiber-reinforced polymers, and potential corrosion when using metal meshes, are thoroughly examined. Solutions such as surface treatments, chemical etching, and the use of nanocomposites are discussed as effective strategies to improve weld quality and mitigate these issues. Additionally, the chapter presents practical applications across various industries, from aerospace to wind energy, underscoring the versatility and growing interest in resistance welding for large-scale and complex structures.

The chapter concludes by underscoring the importance of optimizing welding parameters—such as pressure, heating element type, and processing time—to achieve high-quality welds. The discussed studies demonstrate the successful application of resistance welding in high-performance composites, pointing to its significant potential for broader industrial adoption and the advancement of thermoplastic composite technology.

## 4. Thermoplastic Composite to Thermosetting Composite Joints

Due to the high degree of cross-linking in cured thermoset composites, which hinders the post-production welding of these materials, the common practice for joining fiber-reinforced thermoset (FRTS) materials involves using thermoplastic films or layers as binders or adhesives. This concept can also be applied to fuse thermoplastic–thermoset connections. To achieve this, thermosetting components should be coated with a layer of thermoplastic material before the joining process. Two approaches to this have been proposed: using a chemically compatible co-cured thermoplastic layer or a thermoplastic hybrid intermediate layer [112,113,114,115,116,117].

In designing such connections, it is crucial to recognize that fusion bonding high-performance thermoplastic composites with thermosets presents numerous challenges, primarily due to their differing properties and processing temperatures. While the use of a thermoplastic interlayer does not require chemical and mechanical compatibility (as the interlayer and composite are formed independently), the use of a co-curing layer necessitates careful analysis of the thermoplastic–thermoset interaction [118].

In the co-curing process, a thermoplastic film is applied to the top surface of an uncured composite, and both components are cured simultaneously [118,119,120]. During this process, the initially soluble materials are in contact at temperatures below the glass transition of the thermoplastic, allowing for mutual diffusion. As curing progresses, the molecular weight increases, limiting further diffusion and leading to the formation of a gradient interface through phase separation induced by the reaction.

The literature mentions the use of amorphous thermoplastics, such as polyetherimide, polysulfone, and polyethersulfone, in epoxy composite to enhance adhesion [121,122,123], resistance to crack initiation and propagation [124], and fracture toughness [125]. These thermoplastics are typically soluble in uncured resins and can interact with epoxy matrices in various ways, providing strong interphase bonding after the curing of epoxies [126].

Certain semi-crystalline thermoplastic materials, such as polyamide (PA), may also serve as viable candidates for thermoplastic interlayers, especially when they can interact with or form semi-interpenetrating polymer networks with epoxies [127,128]. This potential arises from their ability to create synergistic composite structures that combine the advantageous properties of both thermoplastics and thermosets, thereby enhancing the overall performance of the joined structure.

Brauner and his team [120] selected suitable thermosetting and thermoplastic materials to produce co-cured composite parts with a thermoplastic boundary layer for subsequent applications. For welding purposes, they chose thermoplastic PEI, determining that its minimum boundary layer should be 80 µm for conventional curing at a temperature of 180 °C.

Research in the literature focuses on resistance welding of thermoset and thermoplastic composites, including the use of thermoplastics as interlayers or binders. This research includes welding carbon fiber-reinforced epoxies with polyetherimide thermoplastic films and binders (such as LOTADEL^®^ AX8900, PARALOID EXL^®^ 2388, and PP-g-MAH). Selected examples are compiled in Table 4.

This chapter explores the challenges and strategies involved in joining fiber-reinforced thermoset (FRTS) materials with thermoplastics. Due to the high degree of cross-linking in thermoset composites, post-production welding is not feasible, making the use of thermoplastic films or layers as binders or adhesives a common practice. Two main approaches for creating thermoplastic-thermoset connections are discussed: using a chemically compatible co-cured thermoplastic layer or a thermoplastic hybrid intermediate layer. The chapter delves into the complexities of these methods, particularly the need for careful analysis of the thermoplastic–thermoset interaction when co-curing. It also highlights the use of various amorphous and semi-crystalline thermoplastics, such as polyetherimide (PEI) and polyamide (PA), which enhance adhesion and improve the mechanical properties of the joints. Research efforts focus on optimizing these techniques to improve the performance and durability of thermoplastic–thermoset welded joints, with examples of successful implementations provided.

## 5. Thermoplastic Composite to Metallic Materials Joints

The use of resistance welding for multi-material joints can be hindered by the differing properties of dissimilar materials, such as thermal conductivity, coefficient of thermal expansion, melting points, and microstructural incompatibilities. Challenges related to welding materials with varying physical, chemical, or mechanical properties, which make it difficult to achieve a solid and durable joint, are known as welding incompatibility.

Scientists are developing numerical models and computer simulations to predict material behavior during welding [129]. These tools aim to optimize process parameters such as time, temperature, and pressure, as well as to forecast and minimize stress.

One approach to avoiding welding incompatibility between different materials is to integrate additional metal elements into the composite as bonding surfaces [130]. The introduction of these connecting elements allows the use of conventional spot-welding guns and the integration of composite components into production lines designed for metals. These components can be incorporated into the composite during or after the manufacturing process.

The literature reports on the development of welding inserts suitable for integration with low-damage thermoplastic composites. Obruch [131] proposed a process where embedding the welding insert requires an additional stage in the process. It has been shown that molded hole joints can carry significantly higher loads compared to drilled samples, provided certain processing conditions are met. However, due to fiber displacement during the forming process, a local material structure is formed, characterized by varying fiber directions [132]. In response, Troschitz et al. suggested an approach where the welding insert is integrated during the manufacturing of the composite component without damaging the fibers [129]. This process is schematically illustrated below (Figure 4).

Nagatsuka et al. [133] addressed the challenge of welding non-conductive composites by using series resistance spot welding (RSW), where electrodes are arranged only on the conductive side of the material. In this method, current is applied exclusively to the metal side, locally heating the metal near the electrodes through resistance heating. The resulting heat causes the carbon fiber-reinforced thermoplastic polymer (CFRTP) matrix near the joint interface to melt by thermal conduction, leading to bonding. According to the authors, the series-RSW method is advantageous for bonding metal to CFRTP, as the electrodes apply pressure to the interface during the heating process, ensuring strong adhesion.

Research in the literature explores resistance welding of thermoplastic composites with metallic materials, such as carbon fiber-reinforced polypropylene with a steel, carbon fiber-reinforced polyamide with aluminum and steel, carbon fiber-reinforced polyetherimide with aluminum, and glass fiber-reinforced polyetherimide with titanium and aluminum. Selected examples are compiled in Table 5.

Another method to overcome the challenges posed by the chemical composition of materials being joined, which hampers their adhesion, involves the appropriate preparation of the surface, particularly the metallic element. Preparation and modifying metal surfaces before joining have several advantages. Surface roughness can increase the bonding area, thereby enhancing adhesion according to the theory of mechanical bonding [141,142,143]. Surface treatment methods can alter surface tension and wettability, both of which are crucial for combining different materials [144,145,146]. In the case of polymers, surface modification is typically performed to increase the proportion of active polar functional groups and enhance the surface energy of the polymer [147].

In the process of welding a fiber-reinforced thermoplastic composite with metallic materials, three primary bonding mechanisms can be identified: chemical bonding, hydrogen bonding, and micromechanical interlocking [148]. These mechanisms can occur simultaneously or independently. The use of special intermediate layers or coatings that react with both the metal and the thermoplastic composite facilitates the formation of connections through chemical or hydrogen bonds. Micromechanical interlocking occurs when the metallic surface is mechanically or additively processed to create microstructures that physically engage with the thermoplastic composite. This method involves precise shaping of the metal surface, enabling the formation of strong micromechanical connections.

In the case of surface treatment applied during resistance welding of metal–composite joints, the study [134] employs standard surface preparation techniques common in adhesive technology. Anodizing with phosphoric acid effectively activates the bonding mechanism between the polymer and the metal, achieved through the mechanical interlocking of PEI with the microporosity of the oxide layer formed on the aluminum substrate.

The impact of surface metal oxide morphology on the initial integrity of metal–polymer bonds in structural applications has been observed [149]. It was found that, in the case of aluminum and titanium, preliminary processing methods such as etching or anodizing can induce the formation of oxide layers on metal surfaces. These oxide layers are characterized by porosity and microscopic roughness, which enhance mechanical interlocking with the polymer. Such interactions lead to significantly stronger bonds compared to those formed on smooth metal surfaces.

Another approach to improving adhesion between metal and composite involves the use of carbon nanotubes (CNTs) in the contact area of the hybrid joint [150]. The optimal LSS of the hybrid joint with a CNT-reinforced plate increased by 146% to 17.28 MPa compared to the hybrid joint without nanotubes.

The scientific literature highlights the use of adhesion promoters, such as silane compounds, to improve adhesion between metal and composite materials [41,42,64,133,139,151,152,153]. These compounds are known for their ability to form covalent bonds with both organic (polymers) and inorganic (metals) materials, thereby enhancing the bonding between these two distinct types of materials. Additionally, the chemical properties of silane compounds allow for the modification of metal surfaces, increasing surface roughness and improving mechanical bonding with polymers. These characteristics make silane compounds effective adhesion promoters in various applications, including the bonding of metals with both thermoplastic and thermosetting composites [154].

Furthermore, it has been reported that the application of a developed silane coating can also prevent moisture absorption at the heating element/polymer interface, which is crucial in preventing corrosion [64,151]. Additionally, it has been observed that polydopamine (PDA), when reacted with a primary amine-containing silane coupling agent (Si-PDA) and deposited on a metallic plate, can significantly delay the degradation of the joint. Si-PDA coatings have been shown to improve bond stability at the titanium–C-PEKK interface due to their effective interaction [155].

This section discusses the challenges and solutions in resistance welding for multi-material joints, focusing on the difficulties posed by the differing properties of dissimilar materials. Various strategies are being explored to address welding incompatibility, including the use of numerical models, metal bonding surfaces, and welding inserts. The text emphasizes the importance of surface preparation, particularly for metallic elements, to enhance adhesion through chemical bonding, hydrogen bonding, and micromechanical interlocking. Methods such as anodizing, etching, and the application of carbon nanotubes and silane compounds are highlighted for their effectiveness in improving bond strength. Additionally, the prevention of corrosion through moisture-resistant coatings like silane and polydopamine is also discussed, showcasing advanced techniques for enhancing the durability of welded joints in composite materials.

## 6. Testing of Welded Joints

The most commonly used method for testing joints is the single-lap shear test, conducted according to ASTM D1002 or ASTM D5868. This test determines the shear strength of a joint during a tension test. The simplicity and widespread use of this method make it effective for comparing welded connections. However, the literature highlights potential limitations of LSS testing [156]. Maximum load or apparent strength values obtained from these tests may be of limited value without a thorough understanding of the joint’s mesostructure, material properties, existing inhomogeneities, and how the mechanics of the joints interact with the test conditions. Therefore, it is recommended to perform additional strength tests, such as critical energy release rate in mode I—double cantilever beam (DCB, ASTM D5528) [157] and short beam strength (SBS, ASTM D2344) [86], to gain a more comprehensive understanding of joint performance

In the recent analyses of joint strength, the focus has predominantly been on assessing static load resistance rather than fatigue loads. However, fatigue loads are crucial, because in real-world applications, welded joints often experience repeated stress over time, which can significantly differ from the conditions observed in standard static strength tests. Fatigue testing is essential for understanding how these joints will behave and potentially fail under continuous use. This is vital for ensuring the reliability and safety of structures that incorporate these welded joints, as it assesses the durability and long-term performance of the joints under cyclic loading conditions [158,159,160].

Another important aspect is the conditions under which tests are conducted. Often, joint tests are performed without accounting for real environmental factors, such as high humidity and temperature. Ensuring the long-term strength and reliability of welds under various operating conditions is crucial. In a study by Villegas [67], the strength of welded joints composed of glass fibers, PPS matrix, and stainless steel mesh was investigated across different temperatures. The study found that joint strength generally decreased as temperatures increased, except between 50 °C and 90 °C, where it remained stable. The primary failure mechanism at all tested temperatures was fiber–matrix debonding, influenced by temperature-related changes in the PPS matrix and stress distribution at the joint interface. The study suggested that temperature-induced variations in residual compressive stress at the fiber–matrix interface significantly impacted overall joint strength.

It has been observed that in titanium–thermoplastic connection [155], water molecules diffusing through the composite layer during exposure to hot and humid environments reach the phase boundary and are primarily adsorbed by titanium. This adsorbed water layer degrades the previously existing interaction between the two adhesives. Conversely, other researchers [161] noted that in the case of resistance welding of CF-PPS, moisture has minimal impact on LSS, regardless of testing temperature and aging conditions. However, it did affect the mode of joint failure.

Currently, there are no standardized procedures for assessing the quality of welds using non-destructive methods. It is generally assumed that such examinations can be conducted similarly to those for adhesive joints in composite structures. Techniques like ultrasonics (e.g., phased array method), infrared thermography and X-radiography method can detect the presence of a welded joint, but they do not provide a quantitative assessment of joint strength [162]. Detecting “kissing bonds”, where surfaces appear well-bonded but lack effective adhesion at the microscopic level, remains particularly challenging, as standard NDT methods may not reveal this defect [163].

Another challenge in non-destructive inspection (NDI) of joints involving different materials is the variation in material properties. In ultrasonic non-destructive testing, for example, the propagation of mechanical waves within a material is influenced by factors such as the material’s elasticity and density, the type of wave (longitudinal or transverse), boundary conditions, geometry, and any heterogeneities in the material structure. These factors directly affect the speed, direction of propagation, and how waves are absorbed or scattered. Developing non-destructive testing methods specifically for resistance-welded joints is essential before this technique can be widely applied in industry.

## 7. Conclusions

This review has provided a comprehensive overview of resistance welding techniques. Recent advancements in resistance welding demand an interdisciplinary approach that integrates knowledge from materials engineering, chemistry, physics, and mechanics. This approach is essential for addressing the complexities of welding different material systems, such as thermoplastic–thermoplastic, thermoplastic–thermoset, and thermoplastic–metal joints. The key findings and conclusions drawn from the literature are as follows:

Effectiveness of resistance welding: Resistance welding is proven to be highly effective in creating strong, durable joints in thermoplastic composites. The technique’s ability to generate localized heat at the weld interface, combined with the appropriate selection of heating elements and process parameters, ensures the successful joining of various composite materials, including complex combinations like thermoplastic and metals.Challenges and mitigation strategies: The review identifies significant challenges in resistance welding, including temperature control, current leakage in carbon-fiber-reinforced polymers, and potential corrosion when using metal meshes. Strategies such as surface treatments, chemical etching, and the integration of nanocomposites have been proposed to overcome these challenges and improve weld quality.Industrial applications and research directions: While resistance welding has been successfully implemented in laboratory settings and smaller-scale industrial applications, such as the Multifunctional Fuselage Demonstrator (MFFD) and Airbus aircraft components, broader industrial adoption requires further research. The challenges of optimizing process conditions—particularly the precise control of energy within the technological window—must be addressed to prevent polymer degradation and fiber displacement. Future research should also focus on the development of reliable non-destructive testing (NDT) methods to ensure the long-term performance and industrial applicability of welded joints.Need for further development: Despite the promising results achieved in laboratory scales and the numerous patents issued, resistance welding still requires substantial research and development before it can be routinely applied to large, industrial-scale structures. A significant challenge lies in optimizing process conditions to balance heat distribution and energy consumption, ensuring high-quality welds without compromising material integrity. This is particularly crucial for metal–composite joints, which present unique challenges due to the differing thermal expansion coefficients and other material properties. The dissimilar nature of these material groups makes it essential to develop specialized approaches that can accommodate their distinct characteristics and ensure robust, durable joints.

## Figures and Tables

**Figure 1 materials-17-04797-f001:**
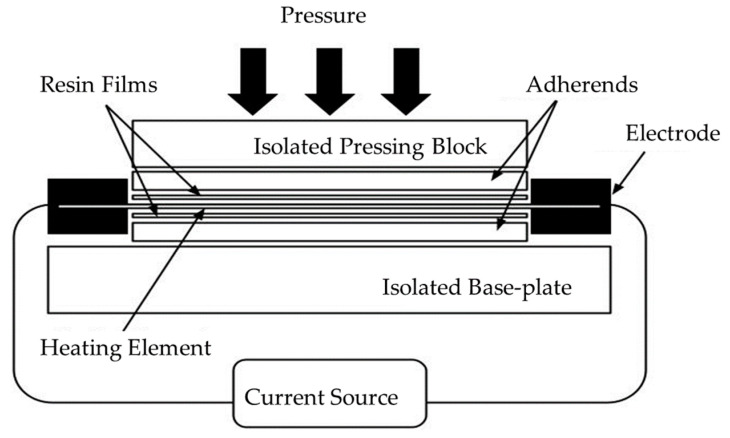
Schematic of resistance welding setup.

**Figure 2 materials-17-04797-f002:**
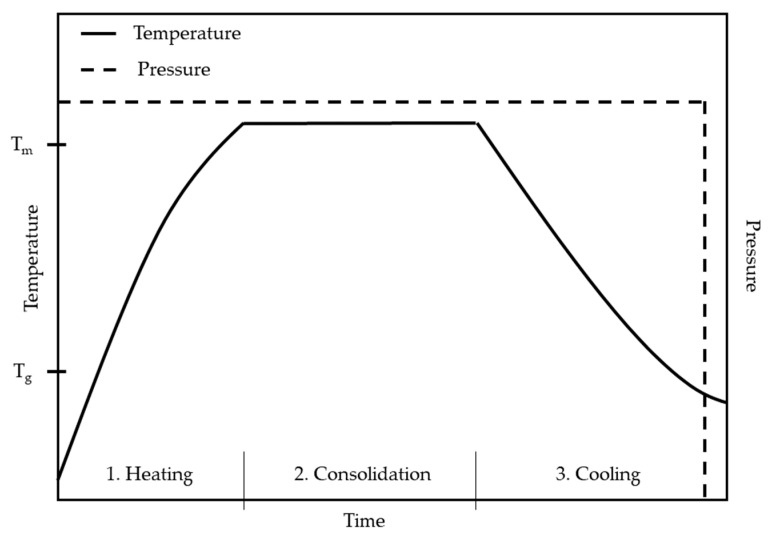
Schematic representation of the resistance welding process for semi-crystalline thermoplastic, where Tm represents melting temperature and T_g_ represents glass transition temperature.

**Figure 3 materials-17-04797-f003:**
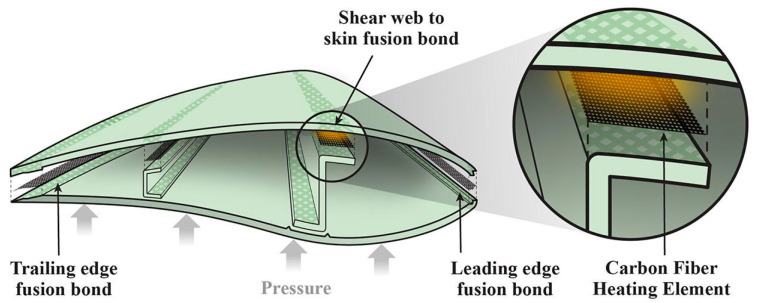
Resistance welding schematic for a wind turbine blade, highlighting the shear web to spar cap bond which is joined by heating the carbon fiber and pressing the mating surfaces together under pressure [111].

**Figure 4 materials-17-04797-f004:**
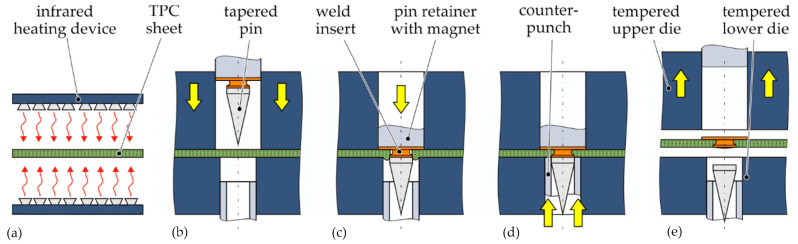
Schematic illustration of process-integrated embedding of weld inserts in thermoplastic composites: (**a**) warming up the TPC sheet (red arrows indicate heat flow), (**b**) closing the compression mould (yellow arrows indicate the direction of mechanical forces), (**c**) shifting forward the pin tool, (**d**) recompressing the squeezed-out material by the counterpunch, (**e**) demoulding of the TPC specimen [129].

**Table 1 materials-17-04797-t001:** Examples of electrical and thermal properties of steel mesh and carbon fibers.

Type of Heating Element	Thermal Conductivity λ [W/(mK)]	Electrical Conductivity σ [S/m]	Thermal Expansion Coefficient α [1/K]	Reference
Metal mesh (e.g., SS304)	16.2	1.39 × 10^6^	17.3 × 10^−6^	[81]
Carbon fibers (PAN *)	7–10	3–8 × 10^−2^	−1.6	[82]

* Polyacrylonitryle (PAN)—precursor for carbon fiber.

**Table 2 materials-17-04797-t002:** Parameters affecting the welding process and the welded joints [83,84].

Process Parameters	Influence	Too High	Too Low
Welding temperature	The optimal temperature ensures proper flow of the material and the formation of a strong weld.	Incomplete fusion, poor connection.	Matrix degradation, air bubbles.
Welding (holding) time	The duration of the process affects the degree of melting of thermoplastic matrix and the interaction between the joined surface.	Incomplete fusion, poor weld.	Overheating, material degradation.
Welding pressure	Appropriate pressure ensures good surface interaction and minimizes the presence of voids in the joint.	Squeezing out the resin, weakening the connection.	Poor adhesion, presence of air voids.
Power input	The input power affects the heating rate of the resistance element.	Insufficient heat, poor connection.	Overheating, burning of the heating element.

**Table 3 materials-17-04797-t003:** Max weld strength of FRTP-FRTP joints with different welding conditions.

Material Combination	Heating Element	Implant Size (Open Gap Width/Wire Diameter)	Time/ Pressure	Parameters Related to Input Energy	Max Weld Strength		References
					**Method**	**Value**	
CF-PPS	SS_MESH_	90 μm/40 μm	60 s/0.7 MPa	110 W	ASTM D1002	37.6 MPa	[64]
CF-PPS	SS_MESH_	—	175 s/0.7 MPa	33.5 A	ASTM D5868	17.1 MPa	[66]
GF-PPS	SS_MESH_	—	300 s/0.7 MPa	30.0 A	ASTM D5868	9.6 MPa	[65]
GF-PPS	SS_MESH_	90 μm/40 μm	55 s/0.8 MPa	80 kW/m^2^	ASTM D1002	13.1 MPa	[67]
GF-PPS	SS_MESH_		60 s/0.8 MPa	80 kW/m^2^	ASTM D1002	24.45 MPa	[68]
CF-PEEK	SS_MESH_	89 μm/40 μm	70–90 s/1.0 MPa	Rising voltage method with an initial voltage of 2.0 V and a rise rate of 9.0 V/min to reach 440 °C (CF-PEEK), 410 °C (CF-PEKK), 390 °C (CF-PEI), 345 °C (GF-PEI)	ASTM D1002	53.0 MPa	[69,70]
CF-PEKK						49.0 MPa	
CF-PEI						45.0 MPa	
GF-PEI						32.0 MPa	
CF-PEI	CF-PEI_PREPREG_	—	—/0.3 MPa	118 kW/m^2^	ASTM D1002	31.0 MPa	[76]
GF-PEI	SS_MESH_	0.16 mm/0.86 mm	150 s/0.2 MPa	20 V/12 A	ASTM D1002	35.44 MPa	[71]
GF-PEI	SS_MESH_	90 μm/40 μm	50 s/0.8 MPa	80 kW/m^2^	ASTM D1002	32.1 MPa	[72]
GF-PEI	SS_MESH_	90 μm/40 μm	55 s/0.8 MPa	80 kW/m^2^	ASTM D1002	34.0 MPa	[108]
CF-PEEK	MWCNT-PEI	10% mass fraction of MWCNTs; d = 10–20 nm; l = 1–12 μm	120 s/1.0 MPa	350 kW/m^2^	ASTM D5868	19.6 MPa	[78]
CF-(LM-PAEK)	CF_T300JB_ with glass fiber insulation	—	I. 30 s II. 15 s	I. 26.6 V II. 18.5 V	DIN-EN 2243-1	~950 N	[74,75]
	CF_T300JB_					~1200 N	
GF-PEI	CNT-SS_MESH_	0.10 mm/0.16 mm	150 s/0.2 MPa	20 V/12 A	ASTM D1002	39.2 MPa	[103]
GF-PEI	GO-PEI	—	90 s/1.5 MPa	340 W	ASTM D1002	39.5 MPa	[104]
CF-(LM-PAEK)	SS_MESH_	40.6 μm/70 μm	-/862 kPa	Weld processing temperature—380 °C	ASTM D3165	45.0 MPa	[73]
CF-PEKK	CNT-PEEK with glass fiber insulation	50 μm	150 s/0.05 MPa	90 kW/m^2^	ASTM D5868	29.0 MPa	[105]

**Table 4 materials-17-04797-t004:** Max weld strength of FRTP-FRTS joints with different welding conditions.

Material Combination	Heating Element	Implant Size (Open Gap Width/Wire Diameter)	Time/Pressure	Parameters Related to Input Energy	Max Weld Strength		References
					**Method**	**Value**	
CF-epoxy/PEI/CF-epoxy	Metal mesh	—	120 s/1.2 MPa	75 kW/m^2^	ASTM D1002	37.5 MPa	[120]
CF-epoxy/CF-LOTADEL^®^/CF-epoxy	CF-LOTADEL^®^ AX8900	d = 0.3 μm	-/0.4 MPa	21 kW/m^2^	ASTM D5528	1879.6 J/m^2^	[119]
CF-epoxy/CF-PARALOID EXL^®^/CF-epoxy	CF-PARALOID EXL^®^ 2388			31 kW/m^2^		1500 J/m^2^	
CF-epoxy/CF-PP-g-MAH/CF-epoxy	CF-PP-g-MAH			31 kW/m^2^		800 J/m^2^	
CF-epoxy/GF-PEI/CF-PEI	CF-PEI	—	3–7 min/0.4 MPa	37 kW/m^2^	ASTM D1022	~20 MPa	[115]
			2–3 min/0.4 MPa	46 kW/m^2^			
			1–2.5 min/0.4 MPa	54 kW/m^2^			

**Table 5 materials-17-04797-t005:** Max weld strength of FRTP–metal joints with different welding conditions.

Material Combination	Heating Element	Implant Size (Open Gap Width/Wire Diameter)	Time/Pressure	Parameters Related to Input Energy	Max Weld Strength		References
					**Method**	**Value**	
CF-PEI/AA7075	CF-PEI with glass fibers insulation	—	10 min/0.4 MPa	90 kW/m^2^	ASTM D1002	25 MPa	[134]
GF-PEI/Ti-6Al-4V	SS_MESH_ with glass fibers insulation	—	120 s/0.2 MPa	20 V, 12 A	ASTM D1002	12.28 MPa	[135]
CF-PA6/SS304	RSW	—	250 s; electrodes were presses on the metal side with the pressing force of 1.5 kN per electrode.	5 kA	Tensile shear test	3.2 kN	[129]
CF-PP/SS304						2.7 kN	
CF-PPS/SS304						unjoined	
CF-PA6/AA6061	RSW	—	70 ms/3600 N	32 kA	Lap shear test	3600 N	[136]
CF-epoxy/GF-PEI/AA6063	SS_MESH_	0.16 mm/0.11 mm	6 min/0.2 MPa	25 V, 20 A	ASTM D1002	19.18 MPa	[137]
CF-epoxy/GF-PEI/AA6063	SS_MESH_	0.16 mm/0.10 mm	4 min/0.2 MPa	25 V, 20 A	Lap shear test	1102.3 N	[138]
CF-PA6/AA5052	RSW	—	0.45 s/2450 N	4400 A	Lap shear test	25.24 MPa	[139]
DC01 */(PA)epoxy	RSW	—	10 ms/3.2 kN	Max 23 kA (for 21 pins)	DIN EN ISO 14273	2656 N	[140]
			10 ms/2.5 kN	Max 20 kA (for 16 pins)		2165 N	

* DC01—mild low-carbon steel for cold forming.

## Data Availability

Not applicable.

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
