# Peer review of "Resistance Welding of Thermoplastic Composites, Including Welding to Thermosets and Metals: A Review"

_materials, 2024, doi:10.3390/ma17194797_

Round 1
Reviewer 1 Report
Comments and Suggestions for Authors
This review paper presents the current progress towards the development of resistance welding of thermoplastic composites. I could not recommend it for acceptance in the present version. it requires significant improvement to be acceptable for publication. Here are the main comments:
1. The structure of the introduction is insufficient. The current introduction fails to provide a detailed description of the issues addressed in this article. The author should consider revising and reorganizing the introduction.
2. Ensure the introduction clearly defines the scope of the study and its significance.
3. The research content in Chapter 2 of the article is very confusing, and there is not much analysis of the principle of resistance welding.The author should consider modifying the title and reorganizing the content of Chapter 2.
4. Use more visual aids, such as graphs, to present research content. This will make the results easier to understand and compare.
5. Tables in the article is too large, it is recommended to adjust it.
6. In the conclusions section, there is a lack of the main conclusions and findings of the paper, and no further research directions or questions have been proposed.
7. Summarize the key findings of the study . Make sure it ties back to the objectives stated in the introduction and abstract.
8. The author's summary and analysis of each chapter in the article are insufficient.
9. Maintain consistency in the format of references.
10. Proofread the manuscript for any grammatical errors and ensure the language is clear and concise.

Comments on the Quality of English LanguageEnglish grammar of this manuscript is acceptable, but some grammar errors may also be found.
Reviewer 2 Report
Comments and Suggestions for Authors
line 58: ... methods are presented are presented in ... -- delete one are presented
line 64-65: two components to be joined - strange space between components and to be
line 69: you are messing up the words insulated in the text and isolated in Figure 1.
line 74: .. as shown in -- referrence number is missing
line 77-80: I don't really understand that sentence. Why is a metall welding mesh a component of the welding equipment and a carbon fibre mesh an additional material to enhance the welding process? You can use a metall mesh or a carbon fibre fabric as welding element. The big advantage of the carbon fibre element is that you are mono-material with the laminates.
line 82-85: The choice between these materials and designs depends on the specific requirements of the welding application, such as the type of the thermoplastic composites, the desired joint strength, and the environmental conditions in which the welded components will be used. --> I would not confirm that. The joint strength does not differ due to the welding element used. Both can be very high. The wish for pure material could be a criterion, for example.
line 85-87: Element made of carbon fiber or metal mesh have different thermal and electrical conductivity properties. --> a metall mesh is a hot-carrier thermistor (NTC) with a negative temperature coefficient and the carbon heating element is a cold-conductor (PTC). You should mention this and explain the difference an what that means for the temperature control.
line 91-29: everal parameters affect the quality and efficiency of welds, including input power, thermal insulation, welding temperature and time, and as mentioned above, the type of heating element --> pressure is also very important! In the table you mentioned it.
Table 2: I think you mixed to high and to low headline. For pressure you definately get squeeze out with high pressure and not with low pressure.
line 239: The most significant project in the field of thermoplastic composites welding is the Airbus-led STUNNING, funded by the Clean Sky 2 program. --> That is not correct. The complete thermoplastic barrel is called MFFD (multifunctional fuselage demonstrator). The Demontsrator was made from two half shells with different technologies. The consortium which built the lower shell was called STUNNING. The Upper shell was build by a consortium from German Aerospace Center (DLR), Premium Aerotec (PAG), Aernnova and Airbus. The two shels where then welded together by Fraunhofer in Stade. More details you can find in the CS article - https://www.compositesworld.com/articles/manufacturing-the-mffd-thermoplastic-composite-fuselage
line 243: The project plans to use resistance welding for joining frames to skin and cleats (shear ties) to stringers - that was used in the Upper shell. By the way the project is already finished and the final demonstrator is now in the ZAL ind Hamburg
l 252: source 58 - the formating of the author of that source in the bibliography should be corrected (Center, I. of S. and D. at the G.A. TB-Rumpf – Technology Bricks for Future Thermoplastic Fuselage 632
Configuration Available online: https://www.dlr.de/bt/en/desktopdefault.aspx/tabid-17959/28476_read- 633
73863/#/gallery/37287 (accessed on 18 January 2024).)
line 255: As shown in the Error! Reference source not 255 found., --> Reference error
line 356: to much space between adhesion and according
line 374: Ageorges --> what is that?
Comments on the Quality of English Language
My comment are included in the Comments and Suggestions for Authors.
Reviewer 3 Report
Comments and Suggestions for Authors
1. As a review paper, the part of the abstract of should provide some important reviewing results and findings.
2. Introduction, it is recommended to add a summary in the first natural paragraph on the performance, cost, advantages and application areas of materials related to this paper, fiber-reinforced thermoplastic and thermosetting resin composites. Some properties and advantages of the two materials are also recommended for comparison and analysis. Some latest research and development should be included, such as https://doi.org/10.1016/j.conbuildmat.2024.136455. https://doi.org/10.54113/j.sust.2024.000038. https://doi.org/10.54113/j.sust.2023.000022.
3. In part 2, during the resistance welding, will some open flame or higher heat be generated during welding? These may have a great influence on the thermoplastic and thermosetting resin composites. Some basic discussion on this point should be paid further attention to.
4. In part 3, for thermoplastic composite to thermoplastic composite joints, simple heating and melting to realize the melting and secondary molding of resin, can realize the reliable connection of the joints. Why do you use resistance welding? It is recommended to provide some relevant explanations.
5. For the resistance welding, what are the labor and electricity consumption costs, welding efficiency, and some trial scenarios? Please make a relevant response to the above comments.
6. For materials with two different softening point temperatures or melting point temperatures, how to consider the relevant resistance welding parameter? Will there be a situation where the welding temperature is too high, resulting in the degradation of the properties of one material?
7. How to ensure the reliable service status of the joints after resistance welding in the service environment? For example, whether some complex environments may lead to corrosion and degradation at the welding joints?
8. What is the current application situation and future development prospect for the resistance welding method? It is suggested to add some related discussions.
9. It is recommended to add some typical review data to the conclusion section.
Round 2
Reviewer 1 Report
Comments and Suggestions for Authors
This review paper presents the current progress towards the development of resistance welding of thermoplastic composites. I recommend accepting it in the current version. The following are the main comments:
1. There is an error in the first line of the third paragraph in Chapter 2.

Author Response
Thank you for your positive recommendation for accepting the manuscript. I also appreciate you pointing out the error in the third paragraph of Chapter 2. The issue has been corrected by removing the bold formatting from the word, ensuring the text is now in its correct form.
Reviewer 3 Report
Comments and Suggestions for Authors
Although the authors provided a revised manuscript, some responses are not still clear. Specific comments are as follows.
l Introduction, for the first comment: “Introduction, it is recommended to add a summary in the first natural paragraph on the performance, cost, advantages and application areas of materials related to this paper, fiber reinforced thermoplastic and thermosetting resin composites. Some properties and advantages of the two materials are also recommended for comparison and analysis”, the authors have added a lot of space to thermoplastic composites, but there is only one reference. However, for the comparison of some performances and advantages, 10 literatures were summarized in one sentence. This expression is unreasonable. As the review paper, it is suggested that the authors consider the first comment and make a detailed analysis and summary of the contents of each part with the reasonable references.
l “Comments 4: In part 3, for thermoplastic composite to thermoplastic composite joints, simple heating and melting to realize the melting and secondary molding of resin, can realize the reliable connection of the joints. Why do you use resistance welding? It is recommended to provide some relevant explanations”, please further explain why the reliable connection between the nodes of thermoplastic composites cannot be achieved by simple heat melting? The adhesive bonding and mechanical fastening may be more suitable for other materials, not thermoplastic composites.
l “Comments 6: For materials with two different softening point temperatures or melting point temperatures, how to consider the relevant resistance welding parameter? Will there be a situation where the welding temperature is too high, resulting in the degradation of the properties of one material? Response 6: This topic was not covered in the manuscript because I did not find any relevant literature addressing the joining of two thermoplastic composites with different polymer matrices. As such, I could not explore the specific considerations for resistance welding parameters in such scenarios. The authors emphasize that there is no relevant document to show the connection of two thermoplastic resins with different melting points. What two kind of thermoplastic composites is applicable to resistance welding? Further clarification is recommended.
l For the Response to Reviewer, relevant responses to comment 9 are missing.
Round 3
Reviewer 3 Report
Comments and Suggestions for Authors
Revised manuscript can be accepted for publication in the current form.